# Trehalose Suppresses Lysosomal Anomalies in Supporting Cells of Oocytes and Maintains Female Fertility

**DOI:** 10.3390/nu14102156

**Published:** 2022-05-22

**Authors:** Woojin Kang, Eri Ishida, Mitsuyoshi Amita, Kuniko Tatsumi, Hitomi Yonezawa, Miku Yohtsu, Daiki Katano, Kae Onozawa, Erika Kaneko, Wakako Iwasaki, Natsuko Naito, Mitsutoshi Yamada, Natsuko Kawano, Mami Miyado, Ban Sato, Hidekazu Saito, Takakazu Saito, Kenji Miyado

**Affiliations:** 1Division of Reproductive Medicine, Center of Maternal-Fetal, Neonatal and Reproductive Medicine, National Center for Child Health and Development, 2-10-1 Okura, Setagaya, Tokyo 157-8535, Japan; ishida-e@ncchd.go.jp (E.I.); amita-m@ncchd.go.jp (M.A.); tatsumi-k@ncchd.go.jp (K.T.); onozawa-k@ncchd.go.jp (K.O.); kaneko-e@ncchd.go.jp (E.K.); iwasaki-w@ncchd.go.jp (W.I.); naito-n@ncchd.go.jp (N.N.); saitou-hi@ncchd.go.jp (H.S.); 2Department of Reproductive Biology, National Research Institute for Child Health and Development, 2-10-1, Okura, Setagaya, Tokyo 157-8535, Japan; ktn990419@gmail.com (D.K.); miyado-k@ncchd.go.jp (K.M.); 3Department of Life Sciences, School of Agriculture, Meiji University, 1-1-1 Higashi-Mita, Kawasaki, Kanagawa 214-8571, Japan; minmin.bb.gd.love.629@gmail.com (H.Y.); ef41579@gmail.com (M.Y.); nkawano@meiji.ac.jp (N.K.); bansato@meiji.ac.jp (B.S.); 4Denentoshi Ladies Clinic, 1-5-1, Azamino, Aobaku, Yokohama, Kanagawa 225-0011, Japan; 5Matsumoto Ladies Clinic, Ikebukuro-Higashiguchi Building F7, 1-41-7, Higashi-Ikebukuro, Toshima, Tokyo 170-0013, Japan; 6Department of Obstetrics and Gynecology, Keio University School of Medicine, 35 Shinanomachi, Shinjuku, Tokyo 160-8582, Japan; mitsutoshi.yamada@gmail.com; 7Department of Food and Nutrition, Beppu University, 82 Kita-Ishigaki, Beppu, Oita 874-8501, Japan; miyado@nm.beppu-u.ac.jp; 8Department of Molecular Endocrinology, National Research Institute for Child Health and Development, 2-10-1 Okura, Setagaya, Tokyo 157-8535, Japan

**Keywords:** oocyte quality, autophagy, chloroquine, trehalose, lysosomal anomalies

## Abstract

Supporting cells of oocytes, i.e., cumulus cells, control oocyte quality, which determines fertilization success. Therefore, the transformation of mature and immature cumulus cells (MCCs and ICCs, respectively) into dysmature cumulus cells (DCCs) with dead characteristics deteriorates oocyte quality. However, the molecular basis for this transformation remains unclear. Here, we explored the link between autophagic decline and cumulus transformation using cumulus cells from patients with infertility, female mice, and human granulosa cell-derived KGN cell lines. When human cumulus cells were labeled with LysoTracker probes, fluorescence corresponding to lysosomes was enhanced in DCCs compared to that in MCCs and ICCs. Similarly, treatment with the autophagy inhibitor chloroquine elevated LysoTracker fluorescence in both mouse cumulus cells and KGN cells, subsequently suppressing ovulation in female mice. Electron microscopy analysis revealed the proliferation of abnormal lysosomes in chloroquine-treated KGN cells. Conversely, the addition of an autophagy inducer, trehalose, suppressed chloroquine-driven problematic lysosomal anomalies and ameliorated ovulation problems. Our results suggest that autophagy maintains the healthy state of the supporting cells of human oocytes by suppressing the formation of lysosomes. Thus, our results provide insights into the therapeutic effects of trehalose on female fertility.

## 1. Introduction

Human ovarian follicles comprise three types of layered somatic cells: theca, granulosa, and cumulus cells, in addition to oocytes [1]. During the maturation of ovarian follicles, granulosa cells undergo differentiation into cumulus cells that surround a single oocyte. Despite their morphological similarities, cumulus cells play a critical role in the formation of fertilization- and development-competent oocytes. Granulosa cell layers are morphologically divided into four types: mature (MCCs), immature (ICCs), dysmature cumulus cells (DCCs), and mural granulosa cells (MGCs) [2].

Autophagy maintains cellular homeostasis by regulating cellular waste clearance. Intracellular vesicles, autophagosomes, enclosed organelles, and parts of the cytoplasm and their contents are degraded upon fusion with lysosomes. LC3, a mammalian homologue of yeast autophagy 8 (ATG8), localizes to the outer membrane of autophagosomes and is a widely used marker for the initiation of autophagosome formation [3]. ATG4 cleaves the LC3 C-terminus as a protease and generates cytosolic LC3-I, which is conjugated to phosphatidylethanolamine and lipidated in a ubiquitin-like reaction in the presence of the E1-like enzyme, ATG7. This lipidated LC3-II binds to the autophagosome membrane [3].

Autophagosomes either fuse with late endosomes to form amphisomes, which eventually fuse with lysosomes, or fuse directly with lysosomes [3]. Lysosomes are organelles that contain more than 50 acid hydrolases as an array of enzymes that digest all types of biological substances, including nucleic acids and proteins [4]. Lysosome-mediated signaling senses the cellular metabolic state and controls the switch between anabolism and catabolism through lysosomal biogenesis and autophagy [4], the mechanism by which the number of lysosomes is regulated is unknown. Autophagy is indispensable for synthesizing new zygote proteins from the resolvents of maternal proteins [5,6].

We recently explored the relationship between cell death and autophagy by examining granulosa cell layers [2]. LC3 proteins were overexpressed in DCCs and MGCs, and chromosomal DNA was highly fragmented in DCCs. However, autophagy initiation was limited to MGCs, as evidenced by the expression of membrane-bound LC3-II and ATG7. Although pro-LC3 accumulated, autophagy was disabled in DCCs, resulting in cell death. Thus, the autophagy-independent accumulation of pro-LC3 proteins leads to cell death. Alternatively, we assumed that autophagic decline may occur in DCCs, consequently mediating the accumulation of pro-LC3. However, the relationship between autophagic decline and cell death remains unclear.

In this study, we explored an event that connects autophagic dysfunction and cell death in the cumulus cells. We determined the effect of chloroquine on changes in size and morphology of lysosomes using two types of cells, mouse cumulus cells, and KGN cells, and examined the ovulation rate using female mice. We further examined the recovery effect of trehalose from suppressed ovulation by chloroquine treatment using female mice.

## 2. Materials and Methods

### 2.1. Reagents and Chemicals

D-(+)-Trehalose dehydrate (trehalose) and chloroquine were supplied by Nacalai Tesque Inc. (Kyoto, Japan). Lysosomes were stained with LysoTracker Green DND-26 (Invitrogen, Molecular probes, Eugene, OR, USA). Cumulus and oocyte nuclei were counterstained with 4′,6-diamidino-2-phenylindole (DAPI; Wako Pure Chemical Industries, Ltd., Osaka, Japan).

### 2.2. Patients and Follicle Stimulation Protocol

This study was approved by the ethical committee of the National Center for Child Health and Development, Japan (#581, 3 July 2012), as described previously [7]. Human granulosa cells (cumulus cells and MGCs) were obtained from patients treated with assisted reproductive technologies (ART) at the National Center for Child Health and Development, Japan. All patients provided informed consent to participate in the study. The patients in the IVF group (*n* = 15) were infertile because of tubal damage or unknown causes. “Unknown” meant that no cause of infertility could be determined by ordinary examinations, including basal body temperature, serum hormones (Luteinizing hormone, follicular-stimulating hormone (FSH), estradiol, or progesterone), hysterosalpingography, ultrasonography, laparoscopy, and semen analysis [8]. Follicular stimulation was performed as previously described [9]. The follicles were stimulated with recombinant FSH (Follistim, Organon, Tokyo, Japan) according to the GnRH agonist long protocol, and ovulation was induced by the administration of 10,000 IU human chorionic gonadotropin (hCG) (Gonadotropin; Aska, Tokyo, Japan) to 18 mm (diameter) leading follicle. Cumulus-oocyte complexes (COCs) were retrieved using an 18-G needle guided by transvaginal sonography from 35 to 36 h after hCG injection.

### 2.3. Cumulus and MGC Preparation

The isolated COCs were washed twice in Sydney IVF fertilization medium (COOK Medical, Brisbane, Australia). Cumulus cells were physically detached from the COCs using 27-G fine needles (Terumo, Kanagawa, Japan). The maturity of COCs was evaluated based on the following morphological criteria: completely expanded cumulus cells with a visible halo (mature), incompletely expanded cumulus cells without a halo (immature), and incompletely or completely expanded and dissociated cumulus cells with dark spots (dysmature). MGCs were isolated from follicular aspirates. For immunofluorescence, a portion of MGCs and cumulus cells were transferred into a dish (Falcon 353001) and washed thrice with 0.1% bovine serum albumin (BSA) in Tris-buffered saline (TBS). The samples were then fixed with 4% formaldehyde in 0.1% bovine serum albumin (BSA) in phosphate-buffered saline (PBS) for 30 min at room temperature, washed three times with 0.1% BSA in PBS, placed on glass slides, air-dried, and stored at 4 °C until use. After detachment from cumulus cells, oocytes were used for ART.

### 2.4. Conventional In Vitro Fertilization (c-IVF) and Intracytoplasmic Sperm Injection (ICSI)

The c-IVF and ICSI were performed as previously described [8]. Briefly, semen samples were collected by masturbation and washed, and motile sperm were separated using a 30–60 min swim-up period. The c-IVF was performed by incubating each oocyte with 50–100 × 10^3^ motile sperms for 5–6 h. For evidence of male factor infertility, ICSI was performed according to a previous report [8]. Oocytes were examined using a dissecting microscope at 16–18 h after insemination or ICSI. The presence of two pronuclei was considered evidence of successful fertilization.

### 2.5. Treatment of Female Mice with Autophagy Inducer and Inhibitor

In the chloroquine-treated group, female C57BL/6J mice (8–12 weeks of age, purchased from Japan SLC Inc., Shizuoka, Japan) were intraperitoneally exposed to 60 mg/kg body weight chloroquine (dissolved in water, 15 μL/g body weight), once every two days, from 6 days before induction of superovulation. In the trehalose-treated group, female C57BL/6J mice were only exposed to chloroquine on day 0 and simultaneously exposed to chloroquine and trehalose (2 g/kg body weight) on days 3 and 6. As reported previously [9,10,11], we decided on the dose of 60 mg/kg in body weight, which effectively inhibited autophagic activity in mice.

To induce superovulation, female C57BL/6J mice (8–12-weeks old; purchased from Japan SLC Inc., Shizuoka, Japan) received intraperitoneal injections of 5 IU (100 μL) of pregnant mare’s serum gonadotropin (Merck4Biosciences, Darmstadt, Germany) followed by 5 IU (100 μL) of human chorionic gonadotropin (hCG; Merck4Biosciences) 48 h later. MII-stage oocytes were collected from the oviducts of females at 14–16 h after hCG administration. The complex of oocytes and cumulus cells (COCs) was then incubated in TYH medium at 37 °C under 5% CO_2_.

All mice were housed (−5 mice/cage) under specific pathogen-free controlled conditions. Food and water were provided ad libitum. All animal experiments were approved by the Institutional Animal Care and Use Committee of the National Research Institute for Child Health and Development (Experimental number A2004-004).

### 2.6. Treatment of KGN Cells with Autophagy Inducer and Inhibitor

The ovarian granulosa cell line KGN [12] was provided by RIKEN BioResource Center (BRC) (Ibaraki, Japan). The cells were cultured in Dulbecco’s modified Eagle’s medium (DMEM) supplemented with 10% fetal bovine albumin (FBS, Gibco BRL, Grand Island, NY, USA), 100 U/mL penicillin, and 100 mg/mL streptomycin, and maintained in a humidified incubator at 37 °C under a 5% CO_2_ atmosphere.

The cytotoxicity of trehalose and chloroquine on KGN cells was detected using the Cell Counting Kit-8 (CCK-8; Dojindo Co., Kumamoto, Japan) assay, according to the manufacturer’s protocol. KGN cells were seeded into 6-cm plates (50,000 cells/cm^2^) and treated with trehalose or chloroquine, as previously described [13]. One day after chloroquine treatment or 3 and 5 days after additional treatment with trehalose, 10 μL of CCK-8 solution was added to each well, and the plate was incubated for 1 h at 37 °C. The absorbance of the wells was measured at 450 nm wavelength using a microplate reader. All experiments were repeated five times.

### 2.7. Lysosomal Observation

To observe lysosomes, human MGCs, cumulus cells, and KGN cells were stained with LysoTracker at a final concentration of 5 nM and incubated for 20 min at 37 °C under 5% CO_2_. The stained cells were immediately observed without fixation.

### 2.8. Electron Microscopic Analysis

KGN cells were fixed with 2% paraformaldehyde and 2% glutaraldehyde in 0.1 M phosphate buffer (pH 7.4) overnight at room temperature. Post which the cells were fixed with 2% glutaraldehyde in 0.1 M phosphate buffer overnight at room temperature and then washed thrice with phosphate buffer (0.1 M) for 30 min. The cells were post-fixed with 2% osmium tetroxide in 0.1 M phosphate buffer at 4 °C for 1 h. The samples were dehydrated in graded ethanol solutions (50, 70, 89, and 100%), transferred to a resin (Quetol-812; Nisshin EM Co., Tokyo, Japan), and polymerized for 48 h at 60 °C. The polymerized resins were ultra-thin sectioned at 70 nm with a diamond knife using an ultramicrotome (Ultracut UCT; Leica, Vienna, Austria) and mounted on copper grids. The sections were stained with 2% uranyl acetate for 15 min at room temperature, washed with distilled water, and subjected to secondary staining with lead stain solution (Sigma-Aldrich) for 3 min at room temperature. The grids were observed under a transmission electron microscope (JEM-1400Plus; JEOL Ltd., Tokyo, Japan). To measure the number of lysosomes categorized into three types (layered, less layered, and shrunken), we counted lysosomes in the cytoplasm of ten cells per experiment.

### 2.9. Statistical Analysis

Significant differences were calculated using the Student’s *t*-test, and statistical significance was set at *p* < 0.05. The results are expressed as the mean ± standard error (SE).

## 3. Results

### 3.1. Characteristics and Outcomes of Patients with Infertility

MGCs and COCs were isolated from the follicular fluid of patients, and the oocytes were separated from the COCs (Figure 1a). The remaining cumulus cells were divided into three types: ICCs, MCCs, and DCCs. We listed the characteristics of the group undergoing conventional in vitro fertilization (c-IVF)-embryo transfer (c-IVF-ET) and/or intracytoplasmic sperm injection (ICSI)-embryo transfer (ICSI-ET) (Appendix A). In fifteen cycles with a single embryo transfer cycle, the mean age of women was 37.5 ± 1.0 years. The mean number of oocytes retrieved from patients was 11.3 ± 1.5, and the fertilization rate was 65.5% ± 4.5%. We categorized COCs obtained from patients by maturation stages and found that the average number was 4.5 ± 1.4 for mature COCs, 4.3 ± 0.6 for immature COCs, and 2.4 ± 0.4 for dysmature COCs.

The percentages of immature oocytes separated from MCCs, ICCs, and DCCs were 6.3% ± 3.4%, 13.5% ± 2.5%, and 38.4% ± 10.4%, respectively (Figure 1b). Immature oocytes surrounded by DCCs had a significantly higher rate than those surrounded by MCCs and ICCs (*p* = 0.019 and *p* = 0.048, respectively). The corresponding fertilization rates of oocytes separated from MCCs, ICCs, and DCCs were 72.7% ± 4.5%, 67.2% ± 6.8%, and 43.9% ± 7.6%, respectively (Figure 1c). Oocytes surrounded by DCCs exhibited a significantly lower fertilization rate than those surrounded by MCCs (*p* = 0.011). This result suggests that the physiological state of cumulus cells is closely related to oocyte maturation, which promises fertilization success.

In turn, to explore the differences between MGCs and DCCs at the molecular level, human mural and cumulus cells were stained with LysoTracker Green DND-26 (LysoTracker) without fixative treatment. The fluorescence signal was the highest in DCCs compared to MCCs and ICCs (Figure 1d). The dysmaturity of cumulus cells is presumably related to the enhanced fluorescence of LysoTracker in the DCCs.

### 3.2. Ovulation Problems Caused by Suppression of Autophagy

Chloroquine is an antimalarial drug known to inhibit autophagic flux by suppressing autophagosome-lysosome fusion [14]. Chloroquine is also the most common drug used to treat acute and chronic inflammatory diseases [15] and is known to decrease the expression of cleaved caspase-3 and suppress apoptosis [16]. As depicted in Figure 2a, this reagent was used as an autophagy inhibitor.

To explore the effect of autophagic dysfunction on female fertility, C57BL/6J female mice were treated with chloroquine (Figure 2b). Female mice were treated thrice with chloroquine and then superovulated with pregnant mare serum gonadotropin (PMSG) and human chorionic gonadotropin (hCG). Subsequently, the number of ovulated oocytes was significantly reduced in mice treated with chloroquine, compared with untreated mice (2.4 ± 1.1 vs. 17.0 ± 1.0; *p* = 0.001).

To explore the effect of chloroquine on lysosomes in cumulus cells, cumulus-oocyte complexes (COCs) were isolated from oviducts of superovulated female mice and stained with LysoTracker. When COCs were treated with 5 μM chloroquine, the fluorescence of LysoTracker was significantly enhanced in cumulus cells, compared with untreated cumulus cells (3.5 ± 0.5 vs. 1.0 ± 0.0; *p* = 0.005) (Figure 2c,d). Based on this result, we assumed that autophagic dysfunction could alter the lysosomal state in cumulus cells, such as their number, structure, and pH.

### 3.3. Lysosomal Anomalies in a Granulosa Cell-Derived Cell Line

To explore the cellular features of the supporting oocytes, we used the KGN cell line derived from steroidogenic human granulosa cells [12]. KGN cells maintain physiological activities, such as the expression of functional FSH receptors, steroidogenesis, and Fas-mediated apoptosis, as observed in human normal granulosa cells [12].

When we observed the percentage of surviving cells following treatment with 0.1 to 0.4 μM chloroquine for 3 days, the number of KGN cells was reduced after 0.3 μM chloroquine treatment (Figure 3a). As the fluorescence signal of LysoTracker increased in MGCs and DCCs, as previously reported [2], we stained the KGN cells with LysoTracker. Corresponding to MGCs and DCCs, the fluorescence signal of LysoTracker increased in KGN cells after 0.2 M chloroquine treatment (Figure 3b), indicating lysosomal proliferation or lysosomal pH alteration. 

### 3.4. Lysosomal Anomalies Caused by Autophagic Dysfunction in KGN Cells

To examine the physiological features of lysosomes after chloroquine treatment, KGN cells were examined by electron microscopy. Lysosomal structures were altered, as they appeared to be structurally heterogeneous, immature, or disrupted after chloroquine treatment (Figure 3c). Based on their structural features, lysosomes were categorized into three types: layered (mature), less-layered, and shrunken (Figure 3d). The percentage of shrunken lysosomes was significantly higher in KGN cells treated with chloroquine than that in untreated cells (26.32% ± 4.60% vs. 0.07% ± 0.01%; *p* < 0.001) (Figure 3e). Moreover, the percentage of enlarged lysosomes was higher in chloroquine-treated KGN cells than that in untreated cells (0.89% ± 0.08% vs. 0.58% ± 0.02%; *p* = 0.017) (Figure 3e). This result indicated that chloroquine alters the lysosomal state, presumably with immature or disrupted features.

### 3.5. Effect of Trehalose on Chloroquine-Treated KGN Cells and Female Mice

As shown in Figure 4a, trehalose is widely considered an autophagy inducer. Trehalose enhances LC3-II expression [17] and induces the nuclear translocation of transcription factor EB (TFEB), a master gene for lysosomal biogenesis, leading to upregulation of TFEB-controlled autophagy gene expression [18,19]. It encodes a transcription factor that coordinates the expression of lysosomal hydrolases, membrane proteins, and genes involved in autophagy [20]. Trehalose treatment counteracts neurotoxicity by upregulating autophagy [21]. Therefore, we examined the effect of trehalose on chloroquine-treated KGN cells. Although chloroquine treatment suppressed cell proliferation in a dose-dependent manner, trehalose reversed the proliferative ability of chloroquine-treated cells in a dose-dependent manner (Figure 4b).

We examined the effects of trehalose on anovulation in chloroquine-treated female mice. C57BL/6J female mice were treated with chloroquine alone or in combination with trehalose (Figure 5a). The mice were then superovulated with PMSG and hCG. To explore the effect of trehalose on lysosomes in cumulus cells, COCs were isolated from the oviducts of superovulated female mice and stained using LysoTracker. In cumulus cells from mice treated with chloroquine alone, the fluorescence of LysoTracker was significantly enhanced, compared with untreated cumulus cells (2.8 ± 0.2 vs. 1.0 ± 0.0; *p* = 0.001) (Figure 5a,b). On the other hand, the chloroquine-driven enhanced LysoTracker fluorescence was significantly reduced by approximately 50% by trehalose treatment (1.8 ± 0.1 vs. 2.8 ± 0.2; *p* = 0.002). Furthermore, the number of ovulated oocytes was significantly reduced in mice treated with chloroquine, compared with untreated mice (5.2 ± 2.1 vs. 16.0 ± 1.0; *p* = 0.013) (Figure 5c). Corresponding to the normalization of enhanced LysoTracker fluorescence, the chloroquine-induced reduction in the number of ovulated oocytes was partially reversed by trehalose treatment (9.7 ± 1.4 vs. 5.2 ± 2.1; *p* = 0.031). These results suggest that trehalose likely plays a suppressive role in chloroquine-driven lysosomal anomalies, leading to an improvement in ovulation disorders. 

## 4. Discussion

The quality of human oocytes that promises fertilization success is associated with the physiological state of cumulus cells. Therefore, physiological determinants of cumulus cells largely contribute to the prediction of oocyte fertility. Here, we propose that autophagic dysfunction is closely linked to lysosomal anomalies (proliferation of shrunken and enlarged lysosomes) and subsequent death of human cumulus cells. We used KGN cells to examine the influence of autophagic dysfunction on cell death in the cumulus cells because it is difficult to obtain large amounts of human ovarian granulosa cells and maintain their primary cultured cells. The KGN cells sustain the expression of follicle-stimulating hormone receptors and participate in steroidogenesis, similar to human ovarian granulosa cells. Similar to DCCs, KGN cells and mouse cumulus cells exhibit lysosomal anomalies by chloroquine treatment. In addition, chloroquine treatment in female mice suppressed their ovulation. Thus, lysosomal anomalies can serve as biomarkers for estimating the quality of human oocytes. Trehalose also reverses lysosomal anomalies and improves female fertility problems. Although these results suggest a possible therapeutic effect of trehalose on female fertility, the clinical use remains speculative and still limited.

### 4.1. Suppression of Autophagy by Chloroquine

Anti-malarial drugs inhibit lysosomal functions [22], and chloroquine is a molecule with hydrophobic characteristics that diffuses into lysosomes, becomes protonated, and gets trapped, thus increasing lysosomal pH [22]. Thus, chloroquine inhibits lysosomal acidification and blocks autophagy [22]. Hence, chloroquine and its derivative hydroxychloroquine are the only drugs clinically used as autophagy inhibitors [22]. These drugs have been tested as anticancer agents in preclinical and clinical trials [23] and in coronavirus disease 2019, shortly, COVID-19 [24]. As shown in Figure 1d, lysosomal aggregation was higher in DCCs than in MCCs or ICCs. Similarly, chloroquine treatment enhanced lysosomal aggregation in KGN cells (Figure 3). In addition, the lysosomal structure was significantly affected by chloroquine treatment (Figure 3c,d), resulting in an altered lysosomal state. If autophagosomes function to remove dysfunctional lysosomes, a reduction in autophagic activity may induce the proliferation of abnormal lysosomes. Thus, we hypothesized that chloroquine-induced autophagic dysfunction could lead to the death of human cumulus cells.

### 4.2. Negative Effect of Chloroquine on Lysosomal Biogenesis

Lysosomal biogenesis increases in response to hypoxic and nutrient-insufficient conditions [25]. Lysosomes also proliferate in malignant cancers [26]. Peroxisomal and lysosomal proliferation is correlated with the activation of enzymes such as catalase and acid phosphatase [27]. As TFEB is responsible for lysosomal biogenesis, a decrease in its nuclear localization and phosphorylated form reduces the number of lysosomes, resulting in autophagic dysfunction [28].

Chloroquine has also been shown to trigger the nuclear translocation of TFEB proteins [29]. TFEB is activated following the exposure of tumors to lysosomotropic drugs, including chloroquine, leading to lysosome-mediated drug resistance in tumors through increased lysosomal biogenesis and drug extrusion via lysosomal exocytosis [30].

### 4.3. Failure of the Clearance of Problematic Lysosomes and Diseases

As high lysosomal numbers are generally observed in advanced malignant tumors, this paradigm may be widely applied to tumors that are resistant to cyclin-dependent kinase (CDK) 4/6 inhibitors [31]. Accordingly, lysosomal cell death may be a plausible treatment for apoptosis-resistant tumors [27,32]. Patients with adult-onset lysosomal proliferation in the liver, termed secondary lysosomal diseases, have genetic defects in lysosomes but not in non-lysosomal organelles [33]. In such patients, the contents and morphologies of proliferating lysosomes are heterogeneous [33]. Lysosomal proliferation during the cytoplasmic remodeling of cultured hepatocytes is a consequence of autophagic activation. Autophagy mediates lysosomal proliferation associated with hepatic remodeling. Therefore, the inhibition of autophagy-driven lysosomal degradation may lead to abnormal protein accumulation in KGN cells through an altered lysosomal state. This phenomenon is likely similar to the lysosomal state of DCCs.

### 4.4. Therapeutic Effect of Trehalose against Human Diseases

As reported in previous [34,35] and current studies [36,37], trehalose is neuroprotective in various animal models of neurodegenerative diseases.

The prevailing hypothesis is that trehalose protects neurons by inducing autophagy, thereby clearing the protein aggregates. Some animal studies have shown activation of autophagy and reduced protein aggregates after trehalose administration in neurodegenerative disease models, seemingly supporting the autophagy induction hypothesis. However, results from cell studies have been less certain. Although many studies claim that trehalose induces autophagy and reduces protein aggregates, these studies have their weaknesses, failing to provide sufficient evidence for autophagy induction theory.

Vertebrates do not synthesize or store trehalose, but retain active hydrolyzing enzyme, trehalase, in the small intestine. Trehalase resides in specific locations, such as the intestinal mucosa, renal brush-border membranes, liver, and possibly the blood. Vertebrates express this enzyme during gestation.

Recent studies demonstrate trehalose has many important properties which show its usefulness in humans [21,38,39]. Non-toxicity of trehalose enables its use in clinical ophthalmology [39,40] and long-term administration in humans [41,42]. Also, trehalose is applied to human oocytes and embryos as a cryoprotectant additive [43]. Despite the various use of trehalose in humans, the risk of its dietary ingestion is debated recently [44,45,46]. It is required to assess the repeatability of its effectiveness and evaluate the safety of its clinical applications.

Our data suggest that chloroquine treatment could impair the ovulation ability of female mice by lysosomal anomalies in the supporting cells of oocytes related to autophagy, which could be rescued, at least in part, by trehalose treatment. A recent study demonstrates that trehalose induces autophagic activation in both porcine cumulus cells and oocytes during oocyte maturation by stimulating two cell type-dependent pathways, suggesting a distinct action of trehalose on oocytes and cumulus cells [47]. To assess the effectiveness of trehalose on cumulus cells, future studies are needed to address whether trehalose contributes to the increased risk of lysosome anomalies and validate the abundance of autophagy-related proteins (LC3 and ATG7).

## 5. Conclusions

As female reproductive processes, including ovulation and pregnancy, are controlled by a series of alterations in hormones, cytokines, and growth factors by environmental and nutritional factors, the examination of a single candidate may be insufficient and lead to misleading conclusions. Overall, our findings highlight the close relationship between autophagic dysfunction and lysosomal anomalies during cell death and may be helpful in predicting the state of cumulus cells that are closely related to oocyte quality.

## Figures and Tables

**Figure 1 nutrients-14-02156-f001:**
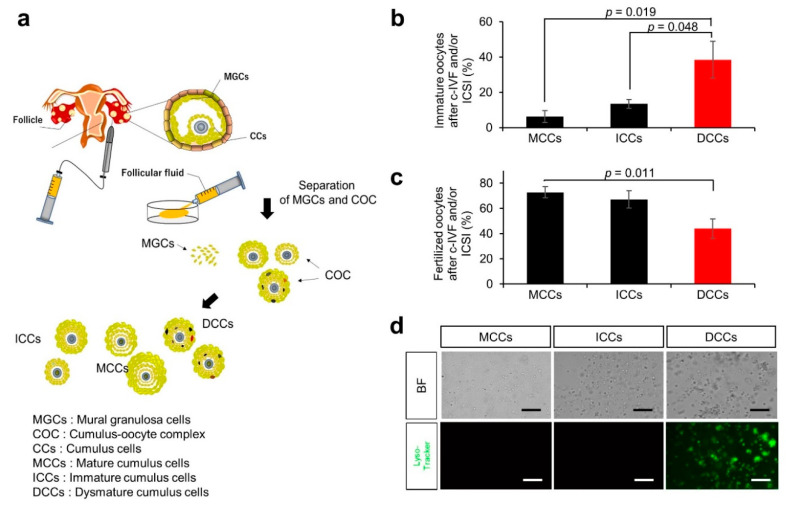
Experimental flow for collecting samples and fertilization rate by the maturity of cumulus cells. (**a**) Flow chart of collecting cumulus cells from cumulus-oocyte complex (COC). COCs were retrieved from stimulated follicles by hormones, and human granulosa cells (MGCs) and COCs were isolated from follicular aspirates. The maturity of COCs was evaluated based on morphologic criteria. Evaluated cumulus cells were physically detached from COCs. Obtained cumulus cells were used in experiments and oocytes were used for ART by IVF or ICSI. (**b**) Percentage of immature oocytes after c-IVF and/or ICSI in oocytes separated from MCCs, ICCs, and DCCs (black bars; MCCs and ICCs, red bar; DCCs). Values are indicated as the mean ± SE. (**c**) Fertilization rate of oocytes after c-IVF and/or ICSI in oocytes separated from MCCs, ICCs, and DCCs (black bars; MCCs and ICCs, red bar; DCCs). Values are indicated as the mean ± SE. (**d**) Staining with LysoTracker. BF, bright field. Scale bar, 50 μm.

**Figure 2 nutrients-14-02156-f002:**
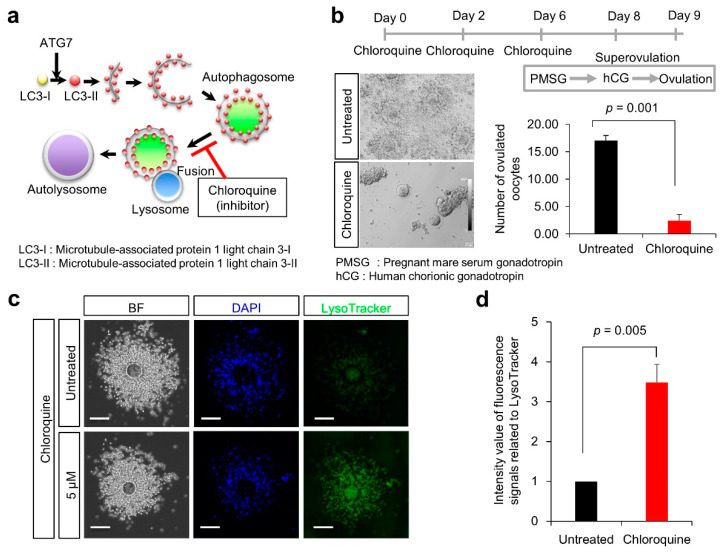
Effect of chloroquine treatment on ovulation rates in female mice. (**a**) Scheme of autophagy and participation of chloroquine in autophagosome-lysosome fusion during autophagy. (**b**) Experimental flow. As depicted in (**b**), chloroquine was injected into 7-week-old female mice (each group: *n* = 5). Ovulated oocytes were counted in superovulation-induced females with (red bar) and without chloroquine (black bar; Untreated). Values are expressed as the mean ± SE. (**c**) Lysosomal distribution. Cells were stained with LysoTracker Green DND26 (green). Scale bars, 100 μm. (**d**) Quantification of fluorescent intensities of lysosomes (over 200 cells were counted in each case). The signal intensity in Untreated (black bar) was arbitrarily set at 1. The graph indicates the relative intensity value of fluorescence signals related to LysoTracker against that in Untreated (black bar). Values are expressed as mean ± SE.

**Figure 3 nutrients-14-02156-f003:**
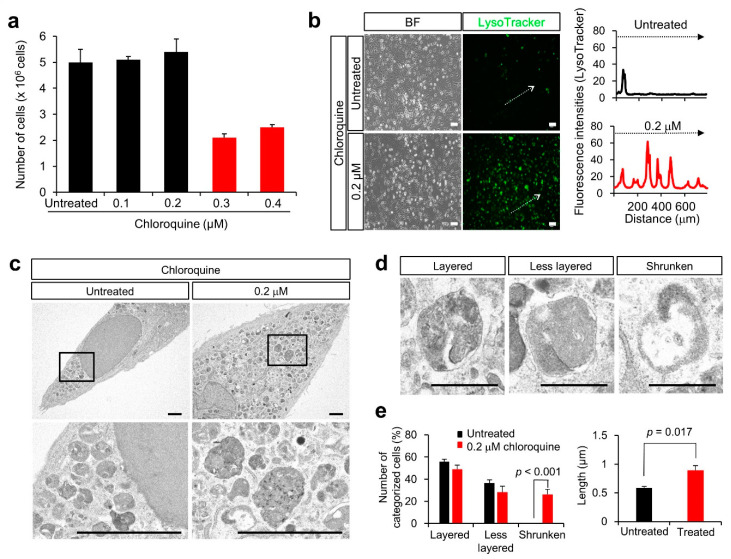
Effect of chloroquine treatment in KGN cells. (**a**) Number of KGN cells after dose-dependent chloroquine treatment (black bars; Untreated, 0.1, and 0.2 μM, red bar; 0.3 and 0.4 μM). Values are indicated as the mean ± SE. (**b**) Staining with LysoTracker and fluorescence intensity. The fluorescence intensity in areas marked with dotted arrows was measured. BF, Bright field. Scale bar, 50 μm. (**c**) Electron microscopic images of KGN cells after chloroquine treatment. Scale bar, 2.5 μm. The boxes are enlarged below. (**d**) Images of lysosomes. The lysosomes were categorized as layered, less layered, and shrunken. Scale bars, 0.5 μm. (**e**) Number of lysosomes is categorized into three groups and measurement of the long axis of lysosomes (black bars; without chloroquine, red bar; with 0.2 μM chloroquine). Values are indicated as the mean ± SE.

**Figure 4 nutrients-14-02156-f004:**
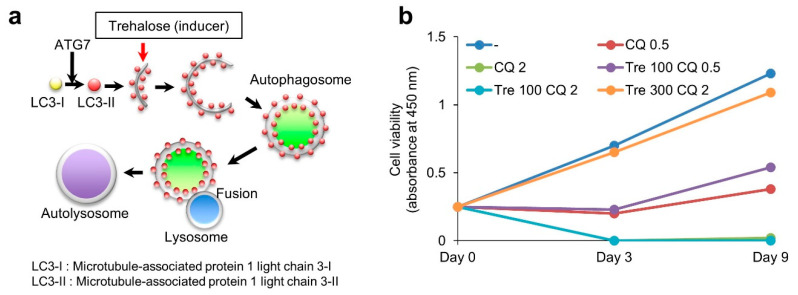
Treatment of KGN cells with trehalose. (**a**) Participation of trehalose (autophagy inducer) in the formation of autophagosome during autophagy. (**b**) Cell viability of KGN cells after chloroquine alone or in combination with trehalose treatment. (-). Untreated. CQ 0.5, chloroquine 0.5 μM treatment. CQ 2, chloroquine 2 μM treatment. Tre 100 CQ 0.5, Trehalose 100 μM + chloroquine 0.5 μM treatment. Tre 100 CQ 2, Trehalose 100 μM + chloroquine 2 μM treatment. Tre 300 CQ 2, Trehalose 300 μM + chloroquine 2 μM treatment.

**Figure 5 nutrients-14-02156-f005:**
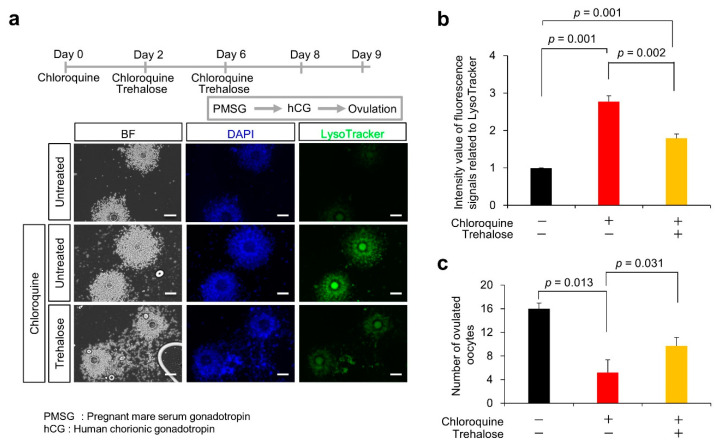
Partial rescue of decrease in chloroquine-induced ovulated oocytes number through trehalose treatment. (**a**) Experimental flow. As depicted in (**a**), chloroquine alone or in combination with trehalose was injected into 7-week-old female mice (each group: *n* = 5). To examine the lysosomal distribution, cells were stained with LysoTracker Green DND26 (green). Scale bars, 100 μm. (**b**) Quantification of fluorescent intensities of lysosomes (over 200 cells were counted in each case). The signal intensity in COC without chloroquine or trehalose treatment (black bar) was arbitrarily set at 1. The graph (red bar; chloroquine alone, yellow bar; in combination with trehalose) indicates the relative intensity value of fluorescence signals related to LysoTracker against that in COC without chloroquine or trehalose treatment (black bar). (**c**) Number of ovulated oocytes after chloroquine alone (red bar) or in combination with trehalose (yellow bar). Black bar indicates number of ovulated oocytes without chloroquine or trehalose treatment. Ovulated oocytes were collected from the superovulation-induced female with and without chloroquine and metaphase II-arrested oocytes were counted. Values are expressed as the mean ± SE.

## Data Availability

Not applicable.

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
