# Peer review of "Trehalose Suppresses Lysosomal Anomalies in Supporting Cells of Oocytes and Maintains Female Fertility"

_nutrients, 2022, doi:10.3390/nu14102156_

Round 1

Reviewer 1 Report

I read with interest this manuscript concerning the sudy of the autophagic process in human cumulus cells. Here some issue to address to improve the manuscript.

  • Although the study design and the results obtained are described in details, the manuscript is very difficult to read. Authors are suggested to make the manuscript more fluent for the reader.
  • Please move the values, or at least in part, reported in the text (results) in a table o figure to make them easier to read
  • Please clarify how the authors selcted the dose for Chloroquine treatment
  • The clinical use of the current finding is speculative and very limited. The authors should discuss this with more caution 

Author Response

May 13, 2022

Manuscript ID: nutrients-1700907, “Trehalose suppresses lysosomal anomalies in supporting cells of oocytes and maintains female fertility”

Dear Editorial board member, Nutrients,

Thank you for your e-mail (May 9, 2022) and the comments from two reviewers concerning the above manuscript. Accordingly, we have improved our manuscript.

Our alterations to the comments are shown below, and our alterations are shown in red in the revised manuscript.

Response to Review

Reviewer#1

Comment 1: I read with interest this manuscript concerning the sudy of the autophagic process in human cumulus cells.

Response: First of all, we appreciate your thoughtful comment.

Comment 2: Here some issue to address to improve the manuscript. Although the study design and the results obtained are described in details, the manuscript is very difficult to read. Authors are suggested to make the manuscript more fluent for the reader. Please move the values, or at least in part, reported in the text (results) in a table o figure to make them easier to read.

Response: Thank you for your valuable comment. We agree to your comment. To simplify the main text, the information regarding the age of patients and the number of oocytes remained, and the sentences including other values were removed (page 4, lines 191-193 in a previous manuscript).

Comment 3: Please clarify how the authors selcted the dose for Chloroquine treatment

Response: We apologize for missing this point and appreciate your comment. According to previous studies, we decided the dose of 60 mg/kg in body weight, which effectively inhibited autophagic activity in mice. We have added the sentence in the materials and methods section (page 3, lines 139-141) and three references (No. 10, 11, and 12).

Comment 4: The clinical use of the current finding is speculative and very limited. The authors should discuss this with more caution 

Response: Thank you for your useful comment. Accordingly, we have added the sentences concerning this point in the discussion section (page 9, lines 338-340, page 10, lines 409-415).

My coauthors and I think that the revised manuscript has been fundamentally improved and that it includes the contents requested by the referees and editorial team.

Thank you for your time and consideration.

Sincerely,

Woojin Kang on behalf of all authors

Department of Reproductive Biology, National Research Institute for Child Health and Development 2-10-1 Okura, Setagaya, Tokyo 157-8535, Japan

Phone: +81-3-5494-7047

Fax: +81-3-5494-7048

Reviewer 2 Report

In this study, the AA aimed to assess if dysmorphic cumulus cells may be responsible for low oocyte quality due to inefficient autophagy.

AA used human granulosa cells from women undergoing ARTs, mouse granulosa cells, and human granulosa cell-derived KGN cell line exposed to chloroquine, an autophagy inhibitor, and to trehalose, an autophagy inducer. Results showed amelioration of ovulation problems exerted by trehalose.

The paper is interesting and innovative but, before possible publication, some aspects should be clarified and better presented, in this referee's opinion:

1)  the aims of the study, especially connected to the use of different models (human, mouse, cell lines) should be better described at the end of the Introduction

2) factors of infertility should be added to M&M

3) approval from the ethical committee should be inserted if needed

4) why AA did not test chloroquine and trehalose also on human GCs?

5) connected to the previous point, it is suggested to better present the pros and the specificity connected to the use of the three used models

6) why EM was performed only on KGN cells?

7) since in a recent paper (not cited, please also evaluate if could be included and discussed) - doi: 10.1093/biolre/ioac060 - GCs and oocyte shows that trehalose activated autophagy by different pathways, it could be of interest to study lysosomes and autophagosomes in mouse oocytes.

Author Response

May 13, 2022

Manuscript ID: nutrients-1700907, “Trehalose suppresses lysosomal anomalies in supporting cells of oocytes and maintains female fertility”

Dear Editorial board member, Nutrients,

Thank you for your e-mail (May 9, 2022) and the comments from two reviewers concerning the above manuscript. Accordingly, we have improved our manuscript.

Our alterations to the comments are shown below, and our alterations are shown in red in the revised manuscript.

Response to Review

Reviewer #2

Comment 1: In this study, the AA aimed to assess if dysmorphic cumulus cells may be responsible for low oocyte quality due to inefficient autophagy. In this study, the AA aimed to assess if dysmorphic cumulus cells may be responsible for low oocyte quality due to inefficient autophagy. AA used human granulosa cells from women undergoing ARTs, mouse granulosa cells, and human granulosa cell-derived KGN cell line exposed to chloroquine, an autophagy inhibitor, and to trehalose, an autophagy inducer. Results showed amelioration of ovulation problems exerted by trehalose. The paper is interesting and innovative but, before possible publication, some aspects should be clarified and better presented, in this referee's opinion:

Response: First of all, we appreciate your thoughtful comment.

Comment 2: the aims of the study, especially connected to the use of different models (human, mouse, cell lines) should be better described at the end of the Introduction

Response: Thank you for your useful comment. We have added the sentences concerning this point in the introduction section (page 2, lines 78-82).

Comment 3: factors of infertility should be added to M&M

Response: Thank you for your suggestion. According to the reviewer’s comment, we have added sentences to the materials and methods section (page 2, lines 97-99; page 3, lines 100-101).

Comment 4: approval from the ethical committee should be inserted if needed

Response: We appreciate your comment. As the reviewer pointed out, we have added the sentence in the materials and methods section (page 2, lines 92-93).

Comment 5: why AA did not test chloroquine and trehalose also on human GCs?

Response: Thank you for your valuable comment. Human GCs were highly fragile and dead in a brief time under the culture condition. Therefore, we considered that human GCs would be not applicable for chloroquine treatment. Instead, since mouse cumulus cells and human KGN cells were resistant to chloroquine treatment, we used these two types of cells.

Comment 6: connected to the previous point, it is suggested to better present the pros and the specificity connected to the use of the three used models

Response: Thank you for your helpful comment. As the reviewer pointed out, we have inserted the sentences to the discussion section (page 9, lines 329-336).

Comment 7: why EM was performed only on KGN cells?

Response: Thank you for your useful comment. Actually, we carried out electron microscopic analysis to examine physiological features of lysosomes in human dysmature cumulus cells (DCCs) (a left figure). Although the DCCs exhibit enlarged lysosomes, the morphological features of lysosomes were diversified among patients. Thus, we excluded the electron microscopic analysis of DCCs from this study.

Comment 8: since in a recent paper (not cited, please also evaluate if could be included and discussed) - doi: 10.1093/biolre/ioac060 - GCs and oocyte shows that trehalose activated autophagy by different pathways, it could be of interest to study lysosomes and autophagosomes in mouse oocytes.

Response: Thank you for your suggestion and recommending a reference. Accordingly, we have inserted the sentences to the discussion section and cited the reference (No. 48) (page 11, lines 418-423).

My coauthors and I think that the revised manuscript has been fundamentally improved and that it includes the contents requested by the referees and editorial team.

Thank you for your time and consideration.

Sincerely,

Woojin Kang on behalf of all authors

Department of Reproductive Biology, National Research Institute for Child Health and Development 2-10-1 Okura, Setagaya, Tokyo 157-8535, Japan

Phone: +81-3-5494-7047

Fax: +81-3-5494-7048

Round 2

Reviewer 1 Report

the manuscrupt gas been improved accordingly